# Revealing spatial-frequency channels in an ensemble encoding model of human fMRI

**Furkan Ozcelik**
New York University
furkan.ozcelik@nyu.edu

**Ajay Subramanian**
New York University
as15003@nyu.edu

**Najib J. Majaj**
New York University
najib.majaj@nyu.edu

**Denis G. Pelli**
New York University
denis.pelli@nyu.edu

## Abstract

Humans recognize objects by means of a narrow octave-wide spatial-frequency filter called a "channel". Behaviorally, this channel can be revealed using critical band masking, a classic psychophysical method that measures sensitivity of recognition accuracy to noise added at each spatial frequency. We want to learn how this channel emerges from the activity of the brain areas involved in object recognition. Critical band masking relies on measured human accuracy in categorizing filtered-noise-perturbed images. To get an analogous accuracy from the brain response, we use N-way Representational Classification Accuracy (N-RCA). For each noise condition used to perturb images in critical band masking, our score measures how often the brain response to the noisy image is more correlated with the response to the original image when compared with responses to $N-1$ other images. This captures how well the brain area's activity informs categorization of a noisy image. We apply the critical band masking paradigm to these accuracies to reveal spatial frequency channels. We then characterize the spatial frequency channel in each visual ROI of an ensemble fMRI encoding model. Our long term goal is to measure channels directly from human fMRI data. Here, we find that the channel bandwidth equals the 1-octave human channel bandwidth and is conserved across model visual ROIs: V1 to V4 and category-selective areas.

## 1   Introduction

Critical band masking, a classic psychophysical method, measures sensitivity of human recognition to noise added in various spatial-frequency bands [1]. Using this method, previous studies found that humans use a narrow, octave-wide band of spatial-frequencies, called a channel, to recognize letters, faces and natural objects [2, 3, 4]. But object recognition is a complex visual process thought to involve several brain areas – early visual areas (V1-V4) and category-selective areas (IT, FFA, PPA, VWFA, etc.). How do the spatial-frequency preferences of these areas contribute to the octave-wide behavioral channel? Behaviorally, critical band masking on each trial asks the human to perform N-way categorization of an image perturbed with noise of some strength (SDs) restricted to one of several spatial-frequency (SF) bands. Categorization accuracy for each noise condition is then used to compute a noise sensitivity threshold for each spatial-frequency. Taken together, these thresholds trace out an octave-wide, inverted-U-shaped curve, the spatial frequency channel (shown in Figure 1). Subramanian et al. [4] used this method to measure behavioral channels in humans and machines performing 16-way ImageNet object recognition. Our goal is to apply their stimuli and method to

38th Conference on Neural Information Processing Systems (NeurIPS 2024).

study spatial-frequency channels in various brain areas. To do so, we need a way to measure a brain area's "categorization accuracy".

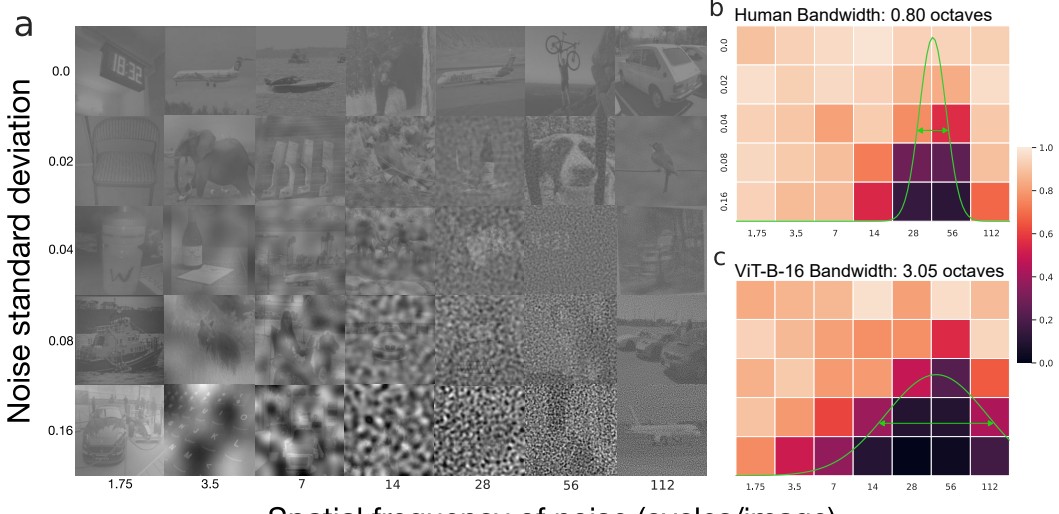

Figure 1: Critical Band Masking (CBM) demo: a) Example images presented in CBM experiment for each pair of spatial frequency of noise and noise standard deviation. b) CBM result of humans on average c) CBM result of ViT-B-16.

To deal with this challenge, we propose a proxy score, called N-way Representational Classification Accuracy (N-RCA) that can be computed using brain responses. Our score (depicted in Figure 2), inspired from pairwise similarity comparison analysis [5], measures how often the representation of the noisy image correlates better with the representation of the original image than those of $N-1$ other noisy images. Thus, N-RCA is analogous to behavioral classification accuracy, and hence can be used to compute sensitivity thresholds of a brain area's response across spatial frequency conditions, which can then be used to derive a channel.

In our analyses, we set $N$ to be 16. This allows us to match chance accuracy $\frac{100}{N}\%$ with chance accuracy in [4]'s behavioral experiments. This new N-RCA score allows us to apply critical band masking to brain responses. This reveals channels in an ensemble of the encoding models of human fMRI data from the Algonauts 2023 challenge [6].

## 1.1 Model selection

To ensure a representative selection, we picked 8 models that spanned diverse training datasets and architectures. We followed Yang et al.'s [8] for training the encoding models. Yang et al. used factorized (in terms of layer, space and scale) and topologically smooth features for encoding. These constraints help the models achieve smoother representations and higher confidence in prediction results. We trained 6 encoding models for the backbone architectures including CLIP [9], DiNOv2 [10], SAM [11], MAE [12], MoCov3 [13] and ViT-B-16 [14] (ImageNet) which exhibited high encoding performance in Yang et al. [8]. We also added two variations of DeiT3[15] architectures trained on ImageNet, DeiT3-B-16 and DeiT3-S-16, due to their low-bandwidth results in behavioral critical band masking analysis [16]. We used data from Subject 1 of the Algonauts challenge which consists of 9841 (image, fMRI trial) pairs. The data was split into 9000 training samples, 200 validation samples (used for ensemble model selection described below), and 641 test samples. All models were trained for 20 epochs. Like [4], we tested our models using grayscale ImageNet validation images reduced to $20\%$ contrast perturbed with Gaussian noise at 4 strengths ($\sigma \in \{.02, .04, .08, .16\}$) and filtered within 7 octave-wide spatial-frequency bands centered at $f \in \{1.75, 3.5, 7, 14, 28, 56, 112\}$ cycles/image. For each model, we collected the encoded representations of layers corresponding to visual areas: V1-V4, and ROIs for Body(EBA, FBA), Face (FFA, OFA), Place (OPA, PPA), and Word (OWFA, VWFA). To consider the best-case encoding model, i.e., an encoding model of human fMRI data that performs better than any single-network encoding model listed above (a closer proxy

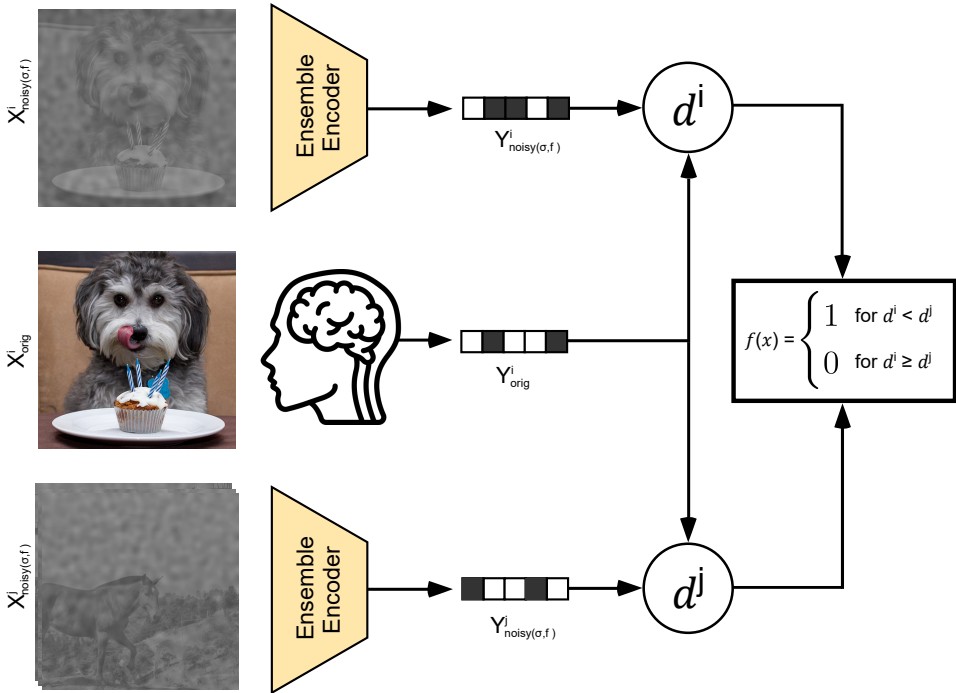

Figure 2: Diagram of N-RCA method. For each noise condition, $(\sigma, f)$, we select $i$th stimulus image $(X^i_{orig})$ and corresponding brain responses $(Y^i_{orig})$ obtained from Natural Scenes Dataset[7] as the ground-truth image and response (we repeat this for every image in the test set), we apply noise to the ground-truth stimulus to obtain corresponding noisy ground-truth image $(X^i_{noisy(\sigma,f)})$ then this image is passed through the Ensemble Encoder to get predicted fMRI response $(Y^i_{noisy(\sigma,f)})$. Then, we select $N-1$ other from the test set (excluding $i$th image) to construct our subset with $N$ images for the comparison (we repeat this for every possible combination of $N$ images in the test set). We apply noise to the every image we have chosen to get noisy comparison images $(X^j_{noisy(\sigma,f)})$, where $j \neq i$) then we pass them through the Ensemble Encoder to get predicted fMRI responses $(Y^j_{noisy(\sigma,f)})$. We compute the correlation distance $(d)$ between groundtruth fMRI response $(Y^i_{orig})$ and predicted fMRI response of the noisy groundtruth image $(Y^i_{noisy(\sigma,f)})$ to obtain $d^i$, and also between groundtruth fMRI response $(Y^i_{orig})$ and predicted fMRI response of the noisy comparison images $(Y^j_{noisy(\sigma,f)})$ to obtain $d^j$. Finally, we compare $d^i$ to all $d^j$ distances in the subset of $N$ samples, if $d^i$ is smaller than all $d^j$ distances then we mark the result as 1, else we mark it as 0. We repeat this computation for all of the possible subsets and get the average score by getting the mean of all subsets which results as our R-NCA score.

to a human fMRI data), we use an ensemble prediction strategy to compute N-RCA. For each voxel in each noise-frequency condition, we pick the encoding model with the lowest L1 distance over the validation set to predict responses for the test set.

We employed the **N-RCA** method to generate accuracy heatmaps for each visual area. We set $N = 16$ with the intention of comparing our results with behavioural channels presented in Subramanian et al.[4]. However, even when a different value of $N$ was used, the results did not vary significantly as long as the threshold was also modified accordingly. 50% accuracy thresholds were then computed for each spatial frequency (after normalizing the scores by the score of original RGB image encoding, which is a ceiling for each ROI) and a Gaussian function was fit to reveal channels.

We observed that for different noise conditions different models stand out with their correlation performances. While CLIP model is best performing model for original color images with 0.257 $R^2$ Score (not presented in CBM experiments), DeiT3-S-16 and DiNOv2 are selected more (in terms of voxels) in the ensemble encoder model for different noise conditions. We also observed that in

every noise condition, ensemble encoder performs better than any individual model. For example, in the case of testing with original color images ensemble encoder model has achieved 0.264 R2 Score. Although it is just a slight difference in the case of color images (which are not included in CBM experiments), it ensured that encoding model to be always better than the best individual performing model for each noise condition.

## 2 Results

### 2.1 Inverted-U-shaped channel

Figure 1 b,c show the octave-wide channel measured by [4] for human behavior, and broader channel measured using N-RCA in one of the deep neural networks, ViT-B-16. Both human and network channels have inverted-U shapes. We characterize human and network channels by 3 properties: bandwidth (log full-width at half max), center frequency (frequency of max. noise sensitivity), and peak noise sensitivity. In this paper, we only focus on bandwidth results since it explains human-machine differences as mentioned in Subramanian et al. [4]

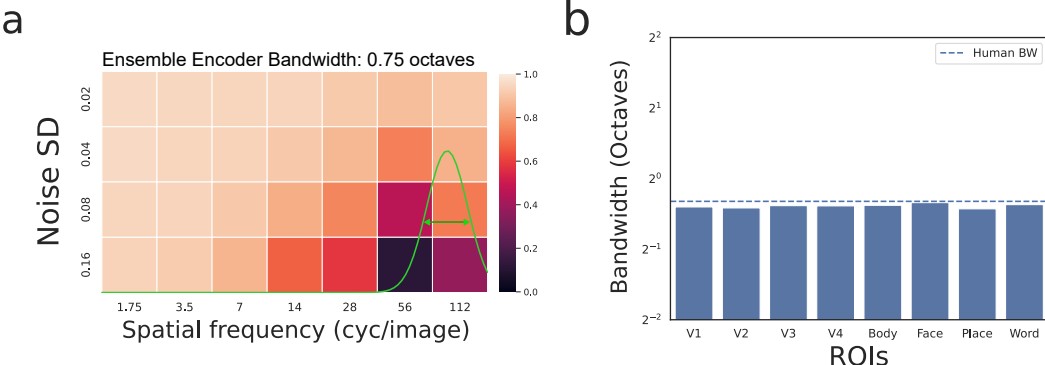

Figure 3: Results of N-RCA for the ensemble encoding model. a) N-RCA CBM result of ensemble encoding model for all the voxels used in Algonauts Challenge. b) ROI-wise bandwidths measured from N-RCA of ensemble encoding model.

### 2.2 Conservation of bandwidth

Subramanian et al.[4] showed that the channel of 76 vision networks, trained to maximize object-recognition performance, is 2-4 times wider than the 1-octave-wide human channel. The larger bandwidth of networks suggests that they use different features for recognition. How do the encoding model for human fMRI data compare?

Fig. 3 plots bandwidths of this ensemble encoding model over several visual ROIs including early visual areas (V1-V4) as well as category-selective areas (bodies, faces, places, words). We see that the channel bandwidth is conserved across the model for all visual ROIs and equals 1-octave, like the human channel. When we examine the ROI-specific bandwidths, in different hypotheses, we could have expected an increasing or decreasing trend, or randomly changing values while we passing from early visual areas to category-specific layers but instead what we observe is a constant channel throughout the ROIs.

## 3 Conclusion

We introduced N-RCA score to critical-band masking to study spatial-frequency preferences across brain areas. We test this method on an ensemble of 8 diverse encoding models of human fMRI data and observe that bandwidth is conserved across all visual ROIs and equal to 1 octave.

As future work, we plan to apply this score to compute channels directly from human fMRI data and compare those with layer-wise channels derived from state of the art deep neural networks.

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
