# OpenReview forum: "Revealing spatial-frequency channels in an ensemble encoding model of human fMRI"
_NeurIPS.cc/2024/Workshop/UniReps — UniReps_

### Official Review · Reviewer_YVsW · 2024-09-29
**The topic and results are quite interesting, but the paper would benefit from additional technical information and more detailed results.**

**Rating:** 6
**Confidence:** 4

**Review:**

The author proposed using N-RCA with critical-band masking to study spatial frequency preferences across different brain areas. They explored several deep neural networks (8 models) with noise-contaminated natural images at various frequencies. They found that all the deep networks exhibited consistent bandwidth across all visual ROIs. The topic and results are quite interesting.

I have several questions:

1. The author mentioned using N-RCA to measure brain responses; however, there isn’t enough detailed information or mathematical explanation to clarify it. While the author does provide some background on N-RCA, the amount of information presented can make it difficult for readers to follow the main topics. I suggest that the authors include a bit more concise information about N-RCA to enhance understanding.

2. I noticed that the author fed noise images into deep neural networks and then measured the predicted fMRI responses, subsequently calculating the correlation between the real and predicted fMRI responses. However, two aspects are not very clear. First, how was the predicted fMRI response measured? Was linear regression or another method used? Second, I would appreciate it if the author could report the correlation values. This information could be included in the appendix if there isn’t enough space in the main text.

3. Following up on the previous question, I wonder about the relationship between physical distance (or Euclidean distance) and perceptual distance (the distance represented in the brain, e.g., neural responses to clean and noise-contaminated images.). By adding noise to clean images, you alter the physical distance (clean vs noise images), which is then fed into deep neural networks. Assuming that these networks perform functions similar to those of the brain, how does physical distance relate to perceptual distance in both the brain and the deep networks?

---

### Official Review · Reviewer_c8mS · 2024-10-04
**Nice but preliminary**

**Rating:** 5
**Confidence:** 2

**Review:**

The paper introduces a metric called N-way Representational Classification Accuracy (N-RCA), which aims to computationally model "channels"—a psychophysical observation that humans use narrow, octave-wide bands of spatial frequencies, known as channels, to recognize letters, faces, and natural objects. The results reveal an inverted U-shaped channel, with the center frequency closely matching human data, but the bandwidth and peak noise sensitivity diverging from human behavior. Despite this difference, the findings are exciting. However, there are several fundamental limitations in the study:

Lack of Methodological Detail: The paper provides insufficient detail about the methodology. The variable "N" in N-RCA likely has a significant impact on the results, yet there is no discussion on its variation or how it affects outcomes. Moreover, the use of eight different encoding models is mentioned, but there is little explanation of how these models were chosen and whether using a different set of models could lead to different results. It is also unclear if all eight models are necessary for the analysis.

fMRI Data and Conceptualization of Noise Response: The study uses fMRI data from general brain responses, but it is uncertain whether fMRI responses would change in the way conceptualized here in reaction to different levels of noise. This raises questions about the applicability of the findings to actual human brain responses.

Stimulus Processing Choices: The method tests grayscale ImageNet validation images reduced to 20% contrast, perturbed with Gaussian noise at four different strengths (σ ∈ {0.02, 0.04, 0.08, 0.16}), and filtered within seven octave-wide spatial-frequency bands centered at frequencies ranging from 1.75 to 112 cycles per image. These values appear to be based on best guesses, without a clear rationale provided for these choices. The lack of justification weakens the robustness of the study’s findings.

Focus on Channel Bandwidth: The paper’s most significant contribution seems to be showing that the model can preserve channel bandwidth to one octave. However, it remains unclear whether optimizing for this behavior has unintended side effects on other model properties. The authors acknowledge the need for further steps, including directly computing channels from human fMRI data and comparing them with layer-wise channels derived from state-of-the-art deep neural networks. Additionally, it is crucial to provide a rationale for model choices and to conduct robustness analyses for these decisions.

---

### Decision · Program_Chairs · 2024-10-10

**Decision:**

Accept

**Comment:**

In light of the reviewers' feedback and relevancy of the submission, we are pleased to accept this paper for presentation at UniReps 2024. We kindly ask the authors to incorporate the reviewers' suggestions and feedback in the final camera-ready version of the manuscript.